# Reasoning Grasping via Multimodal Large Language Model

**Shiyu Jin[1], Jinxuan Xu[1,2], Yutian Lei[1], Liangjun Zhang[1]**
[1]Baidu Research, [2]Rutgers University
reasoning-grasping.github.io

**Abstract:** Despite significant progress in robotic systems for operation within human-centric environments, existing models still heavily rely on explicit human commands to identify and manipulate specific objects. This limits their effectiveness in environments where understanding and acting on implicit human intentions are crucial. In this study, we introduce a novel task: reasoning grasping, where robots need to generate grasp poses based on indirect verbal instructions or intentions. To accomplish this, we propose an end-to-end reasoning grasping model that integrates a multimodal Large Language Model (LLM) with a vision-based robotic grasping framework. In addition, we present the first reasoning grasping benchmark dataset generated from the GraspNet-1 billion, incorporating implicit instructions for object-level and part-level grasping. Our results show that directly integrating CLIP or LLaVA with the grasp detection model performs poorly on the challenging reasoning grasping tasks, while our proposed model demonstrates significantly enhanced performance both in the reasoning grasping benchmark and real-world experiments.

**Keywords:** Robotics Grasping, Multimodal Large Language Model

## 1 Introduction

Robotic grasping has long been a subject of extensive study. The utilization of CNN-based neural networks has demonstrated efficiency in generating high-quality grasping poses from visual input [1, 2, 3, 4, 5, 6]. One significant limitation of these methods is the lack of scene understanding. The robot cannot identify the objects they are grasping. Recent methods have introduced language-guided grasping with instructions such as "I want a stapler" [7] or "Pick the food box in front of the ball" [8]. However, those methods will struggle when dealing with implicit instructions, where the desired object is not explicitly named. For example, if a user says "I need to drink water", an intelligent assistant robot should identify and grasp a cup from a selection of household items, although "cup" does not appear in the user's instruction. Such scenarios are common in real-world settings and the solutions to them remain open questions.

Recently, the advance of large language models like ChatGPT [9] has introduced new possibilities for language reasoning. Beyond text, numerous studies have leveraged multi-modal Large Language Models (LLMs) for visual understanding [10, 11, 12], enabling these models to interpret and generate responses based on both textual and visual inputs. However, these approaches focus on visual comprehension and text generation, lacking the capability to generate robotic actions. While there have been efforts to integrate LLMs with robotics [13, 14, 15, 16, 17, 18], these concentrate on high-level planning.

In this work, we introduce the novel task of Reasoning Grasping (Fig. 1), where a robot determines grasp poses based on users' "implicit" instructions. "Implicit" instructions are language instructions

---

[1]Work done while the authors were with Baidu Research.

8th Conference on Robot Learning (CoRL 2024), Munich, Germany.

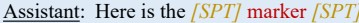
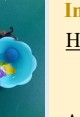
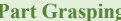
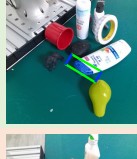
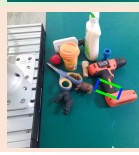

**Explicit Instruction**

Human: *<image>* Pick up the marker. Please output the grasping pose.

Assistant: Here is the *[SPT]* marker *[SPT]*.

**Implicit Instruction**

Human: *<image>* I am hungry and would like to eat. Please output the grasping pose.

Assistant: Here is the *[SPT]* peach *[SPT]*.

**Part Grasping**

Human: *<image>* For the shampoo in the image, which part do you unscrew to open the bottle? Please output the grasping pose.

Assistant: Here is the *[SPT]* cap of shampoo *[SPT]*.

**Multi-round Conversation**

Human: *<image>* What do you see on the table? Is there a power drill?

Assistant: Yes, there is a power drill. It is sitting on top of a workbench along with other items.

Human: What part of this power drill do you grasp to operate it? Please output the grasping pose.

Assistant: Here is the *[SPT]* handle of power drill *[SPT]*.

**Figure 1:** Overview. We integrate the reasoning abilities of multi-modal Large Language Models with robotic grasping. The resulting model interprets complex and implicit instructions, accurately predicts robotic grasping poses for target objects or specific parts within cluttered environments, and supports multi-round conversations with users. In the textual output from the model, the grasping target is indicated by two special tokens *[SPT]*, as demonstrated in the figure. The output grasp poses are visualized in the images with rectangles.

that do not specify the name of the grasp target. A formal definition of reasoning grasping and implicit instruction is given in Sec 3.

To address the reasoning grasping task, we propose a model that directly output grasp poses according to image inputs and language instructions in cluttered environments. Specifically, this model leverages a multi-modal LLM to interpret both images and instructions. We employ a special token strategy to identify tokens that are relevant to the grasping target (i.e., names of target objects). Subsequently, embeddings of these identified tokens are used to generate accurate grasping poses. To train this model, we extend the GraspNet-1 billion dataset [19] with diverse implicit instructions and object part grasping.

Our contribution can be summarized as follows: (1) We introduce a novel task of reasoning grasping, which directs robots to grasp objects based on implicit instructions. (2) We propose a multi-modal LLM for reasoning grasping. Our experiments demonstrate the model's promising performance. (3) We have developed a unique dataset for the reasoning grasping task. It includes 64 objects, 109 parts, 1,730 reasoning instructions, and around 100 million grasping poses. This dataset provides a comprehensive resource for training and evaluating models on reasoning grasping tasks.

## 2  Related Work

**Robotic Grasping.** Robotic grasping has traditionally relied on analytical methods [20, 21, 22], which focus on understanding the geometry of objects or analyzing contact forces. Convolutional neural networks (CNNs) based methods [1, 2, 3, 4, 5, 6] utilize large datasets of labeled grasping examples [23, 24, 19] to train models to predict grasps based on visual input. A critical limitation of both analytical and CNN-based methods is their lack of scene understanding and inability to process language instructions. They do not inherently know what they are grasping, which restricts their application in more dynamic, human-centric environments. Recent advancements in robotic grasping have seen the integration of language understanding, enabling robots to grasp objects based on natural language instructions [25, 26, 27, 28, 29, 30, 31, 32, 33, 34, 35]. This shift allows robots to identify and manipulate objects as specified by users, enhancing their utility in complex scenarios. However, those models require explicit object names as the input and struggle to understand complex and implicit language instructions.

**LLMs for Robotics.** Large Language Models (LLMs) [36, 37, 9, 38] are impressive in understanding and generating human-like text. Multi-modal LLMs can seamlessly integrate with other modalities, such as vision, enhancing their proficiency in tasks like visual understanding [12]. There are also a lot of efforts integrating LLMs with robotics. Many studies [13, 14, 15, 39] have integrated LLMs into closed-loop planning structures, decomposing language-conditioned long-horizon tasks into small steps. Yet, the gap between language instructions and actions still remains. Furthermore, some studies [16, 17, 18, 40] have employed program-like specifications to prompt LLMs, melding planning and action using a predefined library of action functions. There is also an increase in research focused on utilizing LLMs for robotic grasping. Tang et al. proposed GraspGPT to use LLMs to generate semantic knowledge and then selected task-oriented grasp pose from grasp candidates [41]. Mirjalili et al. proposed LAN-grasp to identify the optimal part of an object for grasping with LLM, treating the rest as obstacles, and then applied a traditional grasp planner to determine the best grasp [42]. Compared to GraspGPT and LAN-grasp, which utilize modular frameworks that depend heavily on other pre-trained grasp detection models, our method utilizes an end-to-end training framework and can operate in cluttered scenes.

## 3 Reasoning Grasping

In this section, we define the task of **Reasoning Grasping**, which aims to generate a robotics grasp pose $g$, given an input textual instruction $t$, an input RGB image $v$, and, optionally, an input depth image $v_d$. In the reasoning grasping task, instructions $t$ are not limited to being straightforward, they can also be "implicit". *We define implicit instructions as language instructions that do not explicitly specify the name of the grasp target but provide relevant contextual cues or indicate users' intent implying the grasp target.* For example, implicit instructions may include cues such as shapes, colors, or other attributes of grasp targets, or they may describe users' needs or intent implying the grasp targets based on their functionalities. Some application scenarios are "Users cannot visually confirm the robot's surroundings" or "Tasks that require internal knowledge for decision making". We include the motivations and detailed applications scenarios in Appendix A.

## 4 Reasoning Grasping via Multi-modal LLMs

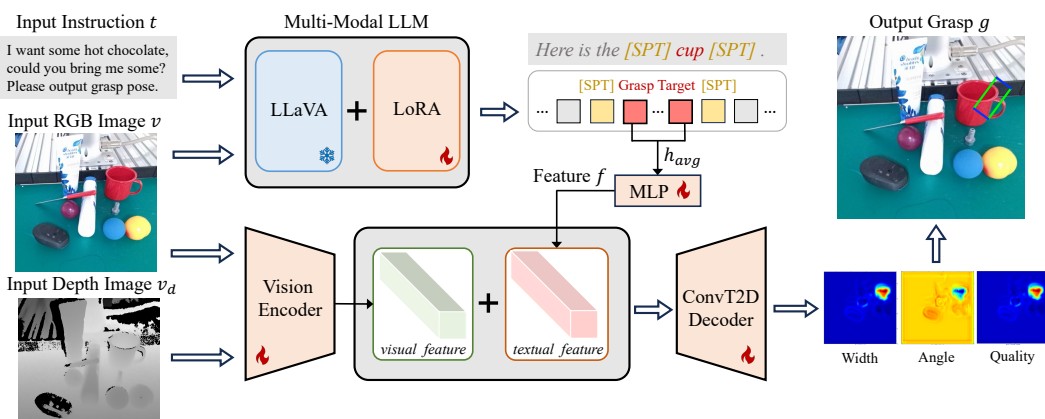

**Figure 2:** Framework of the proposed reasoning grasping model. This model processes visual images $v$ and textual instructions $t$ to output the grasp pose $g$ for the specified target object or part. The embeddings of the grasp target are passed to the grasping module for grasping poses detection.

In this section, we present our model for reasoning grasping that utilizes the reasoning ability of multi-modal LLMs by integrating the extracted features from its textual outputs into the robotic grasping prediction process.

## 4.1 Architecture

The proposed model framework is illustrated in Fig. 2. As for multi-modal LLMs, we utilize the pre-trained LLaVA [12] enhanced with LoRA [43] fine-tuning techniques. LLaVA integrates a CLIP [44] visual encoder with LLaMA [38], an open-source Large Language Model comparable in performance to GPT-3 [37]. And LoRA, known for its computational efficiency, is incorporated into both the projection layer and all linear layers of multi-modal LLMs.

The key role of multi-modal LLMs in the proposed model is to interpret both instruction $t$ and image $v$, then accurately identify the grasping target. Such a target could be an object or, more challenging, a specific part within cluttered scenes. To address the verbosity of LLM outputs, we introduce a special token, i.e., *[SPT]*, strategically placed around the names of the grasping target in the LLM textual outputs. For instance, *[SPT] grasping target name [SPT]*. The purpose of introducing this special token is to identify and isolate the crucial tokens, specifically, the names of the grasping target, from the textual outputs generated by LLMs.

Once the grasping target is identified using the special token, we proceed to extract the last-layer embeddings of its corresponding tokens. It is important to note that there could be multiple tokens associated with the identified grasping target, and we extract embeddings for all identified tokens. These extracted embeddings encapsulate the essential linguistic information related to the grasping target. We then compute the average of these embeddings denoted as $h_{avg}$, which is passed through a projection network to obtain the feature $f$. This feature $f$ is then fed into the image-based grasping detection process, guiding the prediction of the grasp pose.

The grasping detection in our model, rooted in the CNN-based robotic grasping framework in [2], takes n-channel images as inputs. The input image is first passed through three convolutional layers and five residual layers. The extracted visual features are concatenated with the embeddings obtained from the reasoning module. Following this, we use the convolutional transpose operation to up-sample this concatenated feature, increasing its size to match the input image size. The model subsequently generates four output images, each representing different aspects of the grasp pose $g = (x, y, \theta, w, q)$: the grasp quality score $q$, the rotation angle expressed in $\cos 2\theta$ and $\sin 2\theta$, and the required width $w$ for the end-effector. The definition of the grasp pose aligns with previous work [2, 1]. The grasp central coordinate $(x, y)$ is determined from the output image of the grasp quality score. This is obtained by selecting the point in the image with the highest quality score. Notably, the depth image $v_d$ is an optional input modality for the grasping prediction, depending on data availability.

## 4.2 Loss Function

The proposed end-to-end model is trained by a combination of text generation loss $L_{text}$ and grasping prediction loss $L_{grasp}$. The overall objective $L$ is a weighted sum of these two components, formulated as follows:

$$L = \lambda_t L_{text} + \lambda_g L_{grasp}, \tag{1}$$

where $\lambda_t$ and $\lambda_g$ are parameters that determine the relative weights of two loss terms, respectively. Here $L_{text}$ represents the auto-regressive cross-entropy loss for the textual output generated by LLMs, which assesses the performance of identifying the correct grasping target according to input instructions. On the other hand, $L_{grasp}$ evaluates the accuracy of the output grasping predictions, adhering to the loss definition detailed in [2].

## 4.3 Training Strategy

The proposed method leverages the pre-trained LLaVA model, with the LoRA technique for fine-tuning during the training phase. With LoRA fine-tuning, the multi-modal LLM can generate the grasp target within just a few epochs. Based on this observation, our training strategy involves freezing the parameters of the LoRA-enhanced model after the initial epochs. Specifically, from

the third epoch, we set the parameter $\lambda_t$ in the loss function to $0$ and freeze the parameters of the LoRA-enhanced model. Our training then focuses exclusively on other parameters outside the LoRA framework. This approach is designed to reduce the training time and expedite the convergence of the model, particularly in terms of grasping prediction.

Regarding the datasets used for training, our model utilizes both the reasoning grasping dataset (detailed in Section 5) and the original LLaVA Instruct 150K dataset [12] utilized in LLaVA's training. To maintain LLaVA's visual reasoning capabilities, we initially train the model using both datasets concurrently for the first two epochs. Starting from the third epoch, with the LoRA parameters frozen, the model's training proceeds solely with the reasoning grasping dataset.

## 5   Reasoning Grasping Dataset

To train the model for reasoning grasping tasks, we introduce our reasoning grasping dataset, which builds upon the GraspNet-1 billion dataset [19]. GraspNet-1 billion consists of 190 indoor tabletop RGB-D scenes, 88 objects, and 97,280 images. Its wide variety of scenes, objects, and grasp poses provides an excellent foundation. We extend the GraspNet-1B dataset with reasoning instructions as well as object parts grasping annotations. We provide some examples in our dataset in Appendix F.

### 5.1   Reasoning Instruction Generation

We developed a unique and detailed method to create reasoning text instructions for our dataset by creating detailed object descriptions, automated generation with GPT-4, and manual review (Appendix D). The generated instructions enable robots to grasp based on nuanced, context-rich queries.

**Description Creation.** The initial step in our instruction generation process involved creating detailed descriptions for each object and part, providing essential information such as functionalities, physical attributes, and typical uses. For instance, a pair of scissors was detailed as "A handheld cutting instrument with two crossing metal blades pivoted together, typically used for cutting paper or fabric. This particular pair has a black handle with yellow inner grips." These descriptions served as the foundation for generating indirect questions and instructions.

**Automated Generation with GPT-4.** The next step involved generating instructions automatically using GPT-4. An example prompt is provided in Appendix C. GPT-4 was employed to generate indirect questions and instructions for various objects and their parts. The output consists of 10 strings: 5 indirect questions that reference the object's functions or design, and 5 indirect instructions that describe actions involving the object or its usage in certain tasks. These are crafted to be specific, relevant, and direct for grasping tasks.

**Manual Review and Refinement.** Post-generation, each set of GPT-generated instructions underwent a manual review and refinement process. The manual intervention allowed us to fine-tune the language, enhancing the quality and applicability of the instructions for the targeted objects and their parts. This semi-automated approach balanced the efficiency of machine-generated content and judgment of human experts.

### 5.2   Part Annotation

We manually segmented the part point clouds of each object using a specially developed annotation tool. The segmentation process was guided by an understanding of how humans typically interact with and use various parts of an object. For instance, a knife was divided into its blade and handle, reflecting the distinct functional roles of each part. The ground truth grasping poses are assigned to the nearest part. We include the details of part annotation in Appendix E.

### 5.3   Dataset Statistics

In ensuring data quality, low-quality grasping poses were eliminated based on confidence evaluations and objects lacking semantic information were removed. In total, our enhanced dataset com-

prises an extensive collection of 1,730 instruction pairs, 64 objects, 109 segmented parts, and around 100 million associated grasping poses. Unlike other grasping datasets, we offer detailed reasoning instructions and part-specific grasping information.

## 6 Experiments

In this section, we report the experiment results of the proposed method on the reasoning grasping dataset and real-world experiments. In the following experiments, we utilize both explicit instructions, such as "Pick up the <object>", as well as implicit instructions that do not directly mention the grasping target.

### 6.1 Baselines

For the problem of reasoning grasping, we implement two baselines to compare with our proposed model. One baseline employs a CLIP text encoder [44] to extract textual features, while the other utilizes a modular approach incorporating the latest LLaVA model.

**CLIP + GR-ConvNet Baseline.** In this baseline, we combine a CLIP text encoder with a CNN-based grasp detection model, GR-ConvNet [2]. We first utilize the CLIP text encoder to extract the textual feature from input instructions, and this feature is then integrated into the hidden layers of GR-ConvNet. While CLIP can effectively extract language features relevant to visual content, it may fall short in capturing implicit reasoning compared to the VLM.

**LLaVa → GR-ConvNet Baseline.** This is a modular baseline that integrates the latest LLaVA-1.6-34b [45] with a pre-trained grasping detection model GR-ConvNet. As LLaVA cannot directly output the grasping pose, this baseline operates in two stages to obtain the grasping pose: first, LLaVA takes an image and an instruction as the input and is prompted to output a bounding box of the target object in the image. Then GR-ConvNet detects the optimal grasping pose within this specified bounding box. Note that LLaVA-1.6-34b has a much higher performance than LLaVa-v0-7b [12], which is the base model of our proposed reasoning grasping model. This baseline mirrors a similar process to LAN-grasp.

### 6.2 Results

**Text Generation.** We first evaluate the precision of text generation by the fine-tuned multimodal LLM in the reasoning module. Accurately generating special tokens and target names is crucial, since this identified grasping target will be utilized in the subsequent grasping module. We differentiate between explicit instructions, which directly mention the object's name, and implicit instructions, which hint at the target object without stating its name. Note that the instructions used for testing were not seen during the training phase. Results, presented in Table 1, show that our model can successfully interpret 92.52% and 88.79% of implicit instructions for object grasping and part grasping, respectively. This demonstrates our model's good reasoning capabilities in understanding the grasping instructions.

**Table 1:** Text generation accuracy.

|  | Explicit Instruction | Implicit Instruction |
|---|---|---|
| Object Grasping | 99.01% | 92.52% |
| Part Grasping | 95.33% | 88.79% |

**Reasoning Grasping Result.** Table 2 shows the grasp prediction results on the reasoning grasping dataset. Performance is reported using the rectangle evaluation metric [3]. A grasp pose is deemed valid if it fulfills the following two conditions: 1) The Intersection over Union (IoU) score between the predicted and ground truth rectangles exceeds 25%; 2) The angular deviation between the orientations of the predicted and ground truth rectangles is less than 30 degrees. We reported R@$k$, indicating the presence of valid grasps within the top-$k$ grasp predictions. Four distinct scenarios are evaluated, categorized by the combination of instruction types (explicit or implicit) and grasping targets (object or part). We assess both baseline models and our reasoning grasping model with and without depth input. Note that when depth information is utilized in the reasoning grasping model,

**Table 2:** Grasp prediction accuracy on the reasoning grasping dataset.

| Models | Explicit Instruction Object Grasping | | Implicit Instruction Object Grasping | | Explicit Instruction Part Grasping | | Implicit Instruction Part Grasping | |
|---|---|---|---|---|---|---|---|---|
| | R@1 | R@3 | R@1 | R@3 | R@1 | R@3 | R@1 | R@3 |
| LLaVA → GR-ConvNet | 26.80% | 44.66% | 18.34% | 31.29% | 15.38% | 37.93% | 0.92% | 8.53% |
| CLIP + GR-ConvNet | 50.40% | 66.38% | 44.35% | 61.09% | 43.78% | 59.61% | 28.88% | 47.37% |
| Reasoning Grasping | **64.49%** | **86.92%** | **63.55%** | **77.57%** | **59.81%** | **81.31%** | **61.68%** | **83.18%** |
| LLaVA → GR-ConvNet (with depth) | 31.88% | 45.95% | 20.39% | 34.84% | 20.11% | 36.63% | 1.36% | 11.95% |
| CLIP + GR-ConvNet (with depth) | 54.11% | 69.57% | 45.89% | 62.50% | 45.53% | 62.27% | 33.06% | 52.25% |
| Reasoning Grasping (with depth) | **60.75%** | **76.63%** | **57.94%** | **77.46%** | **69.16%** | **77.55%** | **64.48%** | **80.37%** |

it only serves as input for the grasping module and is not incorporated into the reasoning module, as shown in Fig. 2.

The results show that our reasoning grasping model outperforms the LLaVA → GR-ConvNet and CLIP + GR-ConvNet baselines across all scenarios. The first baseline, LLaVA → GR-ConvNet, shows limited effectiveness, as the direct combining of two models fails to convey sufficient information for accurate grasp detection. Although the CLIP + GR-ConvNet baseline exhibits some improvements, it still falls short, particularly in scenarios involving part grasping with implicit instructions. This indicates CLIP's limited capability in processing implicit instructions. In contrast, our model excels in all scenarios, both with and without depth input, showcasing its superior ability to interpret implicit instructions and accurately generate the corresponding grasping poses. Note that some common failure cases are distinguishing objects with similar appearances, such as a shampoo bottle and a hair conditioner bottle, or identifying specific parts of complex objects like the uniformly colored toy camel.

**Visual Reasoning Ability.** Besides reasoning grasping, we also examine the visual reasoning capabilities of our model. The objective is to determine whether the model, after being fine-tuned, maintains the visual comprehension skills of the original LLaVA model. For a quantitative evaluation of performance, we employ a GPT-4 judge to assess the quality of responses generated by our model. This is inspired by the evaluation methods in [12, 46]. According to the judge of GPT-4, the original LLaVA-7B-v0 achieves $8.25/10$ when taking ground-truth textual descriptions as references. On the other hand, our model achieves $7.43/10$ under the same GPT-4 judge. This shows that our fine-tuned model still preserves a good portion of the reasoning ability of the original LLaVA. We include the details of the experiments in Appendix J.

### 6.3 Real World Experiments

We conduct real-world experiments to evaluate the proposed reasoning grasping model and the CLIP + GR-ConvNet baseline. A 7-DoF Franka Emika Panda equipped with a Franka Hand parallel gripper was utilized for grasping execution. To perceive the objects, one Azure Kinect was positioned in front of the robot. A cropped $480 \times 480$ RGB image of the workspace was utilized as the input to the model. To execute the 2D grasps generated from the model, we always apply a top-down grasp with the height computed using the depth of the grasping point. In our experiments, we randomly place 4-8

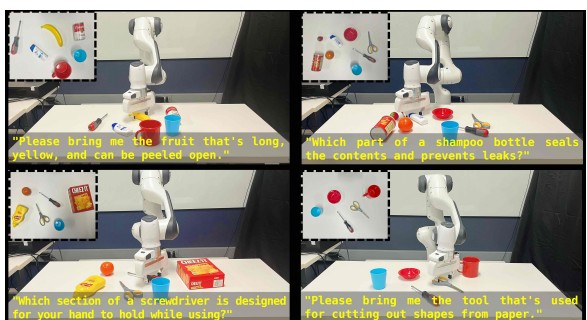

**Figure 3:** Real world experiments with implicit instructions.

objects on a tabletop and provide either explicit or implicit language instruction to the robot to grasp an object or a specific part, as shown in Fig. 3. We select objects that are either identical or sim-

ilar to those in the dataset. We deliberately excluded any objects or parts that are not infeasible for grasping, such as items too wide for the robot's gripper. The performance is evaluated using three evaluation metrics: the correctness of generating special tokens and grasping target names, the accuracy of the output grasp pose, and the success in lifting the object. The results are reported in Table 3.

Overall our proposed model can generate accurate special tokens and target names 39 out of 40 trials. This again shows the excellent reasoning capabilities of our model to interpret the various instructions. The failure

**Table 3:** Real-world experiment results.

| | Model | Token Accuracy | Pose Accuracy | Execution Success |
|---|---|---|---|---|
| Object / Explicit | CLIP + GR-ConvNet | N/A | 5/10 | 2/10 |
| | Reasoning Grasping | 10/10 | 7/10 | 5/10 |
| Object / Implicit | CLIP + GR-ConvNet | N/A | 3/10 | 2/10 |
| | Reasoning Grasping | 9/10 | 5/10 | 3/10 |
| Part / Explicit | CLIP + GR-ConvNet | N/A | 5/10 | 3/10 |
| | Reasoning Grasping | 10/10 | 7/10 | 4/10 |
| Part / Implicit | CLIP + GR-ConvNet | N/A | 2/10 | 2/10 |
| | Reasoning Grasping | 10/10 | 6/10 | 4/10 |

case involved the model incorrectly identifying a screwdriver instead of scissors in response to a prompt requesting a tool to open a package box. We classified it as a failure solely because it did not match the specific pairing in our dataset, which serves as the basis for our evaluation metrics. This example underscores that even when the model's response is technically incorrect, it is still reasonable. In terms of grasping performance, we assessed top-3 output grasp poses and manually selected the most feasible one for execution on the robot arm. We conducted 10 trials for each scenario, totaling 80 trials, to compare reasoning grasping with CLIP + GR-ConvNet baseline. The CLIP + GR-ConvNet baseline and our reasoning grasping model produced 15/40 and 25/40 accurate grasping poses, respectively, with the success rates for lifting objects are 9/40 and 16/40, respectively. The main reasons for accurate poses but fail to lift were due to the object slipping from the gripper or the object being too heavy. The experiment results demonstrate that our model outperforms the baseline in four distinct scenarios, showing its superior ability in reasoning grasping tasks. It's important to emphasize the unique challenges posed by our study, which involve "implicit" instructions, grasping in cluttered environments, and task-oriented objects/parts grasping. This level of complexity has not been studied before. As a comparison, a recent study [8] deals with grasping in cluttered environments with "explicit" instructions achieves $60\%$ for grounding accuracy (accurate poses) and $20\%$ for success rate (success lift), respectively.

## 7   Conclusion

In this paper, we introduce a novel task of reasoning grasping, where robots need to interpret and generate grasping poses based on implicit instructions. By leveraging the reasoning ability in a multi-modal large language model, our proposed model can understand indirect verbal commands and generate corresponding grasping poses. In addition, we also present the first dataset for reasoning grasping, derived from the GraspNet-1 billion dataset, featuring implicit human instructions alongside object and part grasping annotations. Experiment results demonstrate that our approach can effectively comprehend implicit instructions and accurately generate corresponding grasping poses. Overall, this work offers valuable insights bridging implicit human instructions and generating precise robot actions.

While the proposed reasoning grasping model shows promising results, the model is limited when generating optimal grasping poses for novel objects and refining poses through conversational interactions. A possible reason is that the variety of objects with grasping annotations is still limited. For future work, the LLaVA-v0-7b model is not the state-of-the-art Visual Language Model. Other models such as LLaVA-1.6 [45] or GPT-4V [47] have shown much better performance and stronger ability in terms of visual understanding. Our model's architecture, with its modular reasoning and grasping components, allows for flexibility in upgrading to more sophisticated models in the future.

**Acknowledgments**

We would like to express our gratitude to the reviewers for their insightful feedback and constructive comments. Additionally, we thank Zhixian Ye for his assistance with the experiments during the paper rebuttal.

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

## A    Motivations and Application Scenarios

We aim to advance robotic systems to not only understand direct human commands but to also possess the reasoning capability to interpret implicit instructions. These implicit instructions can provide a general description of users' needs, requiring the robot to infer the grasping target on its own, without explicitly naming the object. This is very common in real-world scenarios. Here are several practical application scenarios where direct instructions may be unavailable:

- **Users cannot visually confirm the robot's surroundings:** For example, blind people can instruct the robot to "find a sweater in a warm color that matches the pants". The robot could also be asked to "find something to hold these papers together", where it might choose from a stapler, paper clips, or binder clips based on availability.

- **Tasks that require internal knowledge for decision making:** Robots can utilize their internal knowledge for tasks like sorting in recycling facilities. They could be instructed to "sort materials that are recyclable and compostable", identifying items based on texture and material type, which facilitates sorting without the need to know whether the objects are recyclable or not.

- **Naming objects is impractical or complex:** In some situations, explicitly naming objects is impractical or the names are complex and difficult to remember. For instance, telling a robot, "I need my morning medicine", allows the robot to use its routine knowledge to fetch the correct medication without the user needing to recall specific drug names, which are usually long and complex.

- **Managing multiple items:** Explicitly naming multiple objects can be cumbersome and inefficient. In emergency medical situations, a medical robot might be instructed to "gather all items necessary for suturing wounds". It would need to discern which tools and supplies are relevant, such as needles, thread, and antiseptics, based on the medical context provided. For general tidying tasks in home organization, a home robot could be instructed with phrases like "The living room is messy, tidy up by picking up toys", allowing the robot to identify and collect toys without naming each item individually (e.g., teddy bear, toy train, etc.).

## B    Compared to Other Methods.

Here we provide detailed comparisons with similar approaches: GraspGPT [41], LAN-grasp [42], and GPT-4v [48].

### B.1    Compared to GraspGPT and LAN-grasp.

**Clutted scenes.** GraspGPT and LAN-grasp are designed to grasp specific parts of single objects, whereas our method focuses on grasping in cluttered scenes. During our experiments, we intentionally tested our model in cluttered scenes to assess its ability to reason about grasping targets. These targets could be entire objects or, more challenging, specific parts within cluttered scenes.

**End-to-end trained.** One significant distinction is that our model is an end-to-end system for directly generating numerical grasp predictions. However, GraspGPT and LAN-grasp have modular frameworks that both employ external pre-trained models for grasp detection.

- GraspGPT, illustrated in Figure 4, has to take grasp candidates as the input. GraspGPT serves as a grasp evaluator, selecting a grasp pose from given input grasp candidates, which are generated from pre-trained models. This reliance on frozen models designed for single objects limits GraspGPT's adaptability to complex environments, such as cluttered scenes.

- LAN-grasp generates the bounding boxes of targets using Vision-Language Models (e.g., OWL-Vit [49]) and then uses GraspIt! [50] for determining the output grasp pose. This

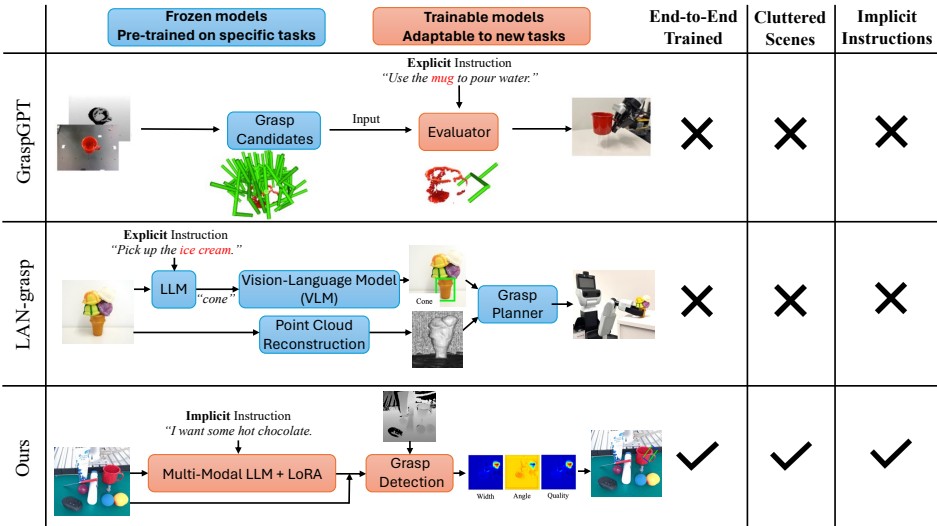

**Figure 4:** Comparison of our method with GraspGPT and LAN-grasp.

process without any trainable model heavily depends on the performance of utilized pre-trained models. Also, it is challenging to adapt LAN-grasp to new tasks or scenes. More-over, our baseline"LLaVA → GR-ConvNet" mirrors a similar process to LAN-grasp, uti-lizing LLaVA [12] for bounding box identification and a pre-trained GR-ConvNet [2] for grasp pose detection. However, its performance is unsatisfactory when applied to cluttered scenes on our reasoning-grasping dataset, as shown in Table III of the paper. In contrast, our model jointly trains language models and grasping detection models, eliminating the need for external pre-trained models. Experimental results (Table III) demonstrate that our end-to-end approach outperforms modular frameworks such as "LLaVA → GR-ConvNet".

## B.2   Compared to GPT-4v.

While models like GPT-4 with vision (GPT-4v) [48] show potential in tasks such as object detection, they still face challenges in new tasks with generating unique numerical predictions like grasp poses, despite careful prompting. To illustrate GPT-4v's ability to generate numerical predictions for novel tasks like robotic grasping, we conduct additional experiments using both zero-shot and few-shot prompting techniques. Specifically, we use images from the benchmark Cornell Grasp dataset [4], where each image contains only a single object. GPT-4v is prompted to generate grasp poses for the object in the scene. As shown in Figure 5, adapting GPT-4v to new grasp detection tasks, even in simple scenarios with single objects, remains challenging despite careful prompt design and providing examples.

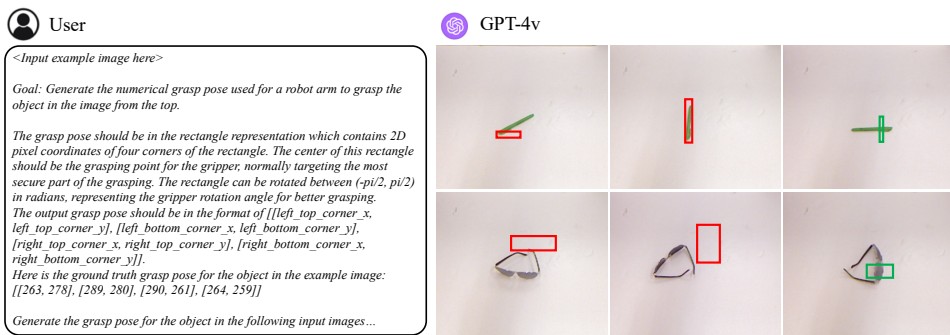

**Figure 5:** Grasping detection with few-shot prompting using GPT-4v on Cornell Grasp dataset.

## C Prompt for Instruction Generation

```
        System Prompt for Instruction Generation

You are tasked with creating specific, indirect questions and instructions that a robot
could use to identify and interact with objects based on their names or detailed
descriptions provided by users. When an object is given, such as a pair of scissors
described as "A handheld cutting instrument with two crossing metal blades pivoted
together, typically used for cutting paper or fabric. This particular pair has a black
handle with yellow inner grips," you must formulate responses that precisely hint at the
object's uses or features without naming it directly. The aim is to enable the robot to
deduce the correct object through these indirect cues, enhancing its ability to understand
and execute tasks involving the object.

Your output should include:
· Indirect Questions (5 Total): Construct questions that indirectly reference the object's
  specific functions, design attributes, or contexts in which it is used. These questions
  should guide the robot to consider the essential features or tasks the object is
  associated with, aiding in its identification without directly mentioning the object's
  name.
· Indirect Instructions (5 Total): Develop instructions that subtly describe actions
  involving the object or request its utilization for particular tasks. These instructions
  should be carefully phrased to convey the object's use or physical characteristics
  indirectly, allowing the robot to infer which object is needed without explicit naming.

Please format your responses as a list of 10 strings, organized as follows: ['Indirect
Question 1', 'Indirect Question 2', ..., 'Indirect Question 5', 'Indirect Instruction 1',
..., 'Indirect Instruction 5'].

In structuring your responses, prioritize:
· Specificity and Relevance: Use language that precisely hints at the object's attributes
  or uses, ensuring the robot can accurately identify and grasp the intended object.
· Directness and Functionality: While maintaining indirectness, your hints should be
  straightforward and functional, focusing on enabling the robot to understand and act
  upon the instructions or questions effectively.
· Consistency and Clarity: Ensure each hint is consistently structured and clear, avoiding
  overly creative or ambiguous phrasing that could confuse the robot's learning process.

This methodical approach is designed to improve the robot's ability to interpret indirect
language and identify objects based on functional cues, thereby enhancing its interaction
with and manipulation of objects in its environment.
```

**Figure 6:** Example Prompts used for Initial Reasoning Instruction Generation with GPT-4

# D Reasoning Instruction Generation

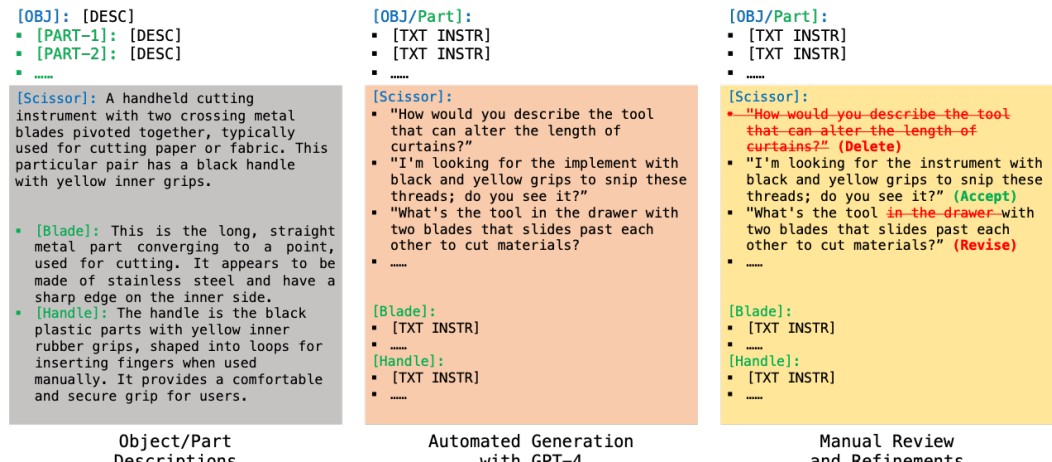

**Figure 7:** Reasoning instruction generation. The instructions generation process involves 1) Object/Part Descriptions; 2) Automated Generation with GPT-4; and 3) Manual Review and Refinements.

# E Part Segmentation and Grasping Pose Annotation

Our dataset's annotation process includes two key components: pose assignment and pose projection, for both 3D and 2D grasping poses, as shown in Fig. 8. Pose assignment involves aligning the grasping poses with the nearest part of the object. For each object's segmented parts, we determine the grasping center of each point. These grasping poses are then assigned to the nearest part, ensuring an accurate association with the specific object part they correspond to. Pose projection transforms the grasping poses of parts into both 3D and 2D representations. This is achieved using camera matrices and object transformations. In the case of 3D, the original grasping poses are preserved in the spatial domain. For 2D projection, the 3D points of each segmented part are projected onto 2D images. This results in 2D masks that depict the spatial distribution of each part on the image plane. Grasping poses within these masks are then mapped to the corresponding parts in 2D.

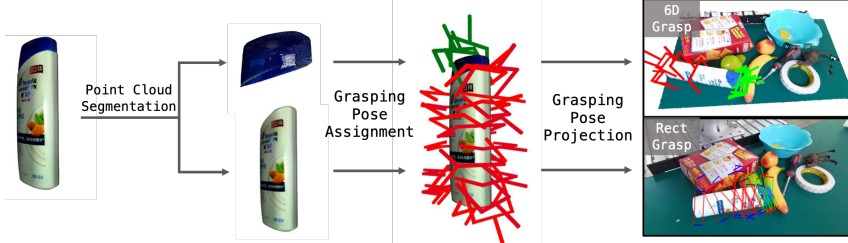

**Figure 8:** Part grasping annotation process: 1) the object's point cloud is manually segmented into distinct parts; 2) annotated grasping poses are allocated to each segmented part of the object; and 3) the grasping pose is projected to 2D and 6D.

# F Examples of Reasoning Grasping Dataset

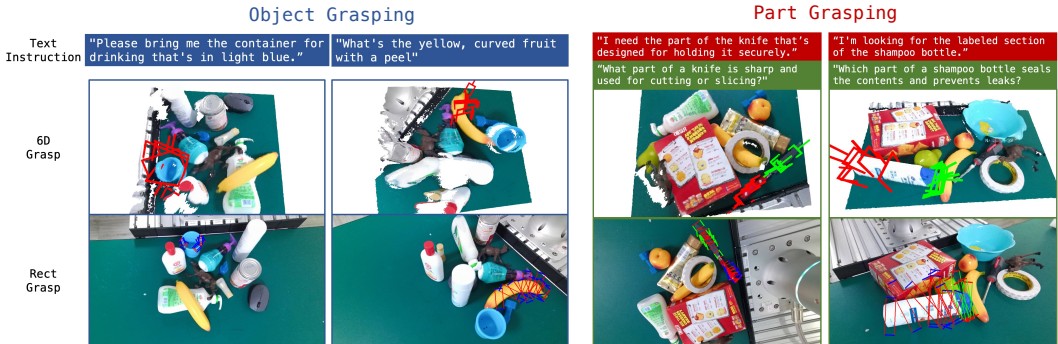

**Figure 9:** Our reasoning grasping dataset enriches the GraspNet-1 billion dataset [19] with additional annotations for part-level grasping and reasoning instructions.

# G Comparison with Different Datasets

**Table 4:** Comparison with different datasets

| | Grasp Label | Modality | Multi Object | Num. Objects | Grasps per Object | Num. Grasps | Num. Samples | Part Annotations | Reasoning Instructions |
|---|---|---|---|---|---|---|---|---|---|
| LISA[51] | ✗ | RGB | ✔ | N/A | N/A | N/A | 1,218 | ✗ | ✔ |
| DectGPT [52] | ✗ | RGB | ✔ | N/A | N/A | N/A | 5,000 | ✗ | ✔ |
| Cornell [23] | Rect. | RGB-D | ✗ | 240 | 33 | 8,019 | 1,035 | ✗ | ✗ |
| Jacquard [24] | Rect. | RGB-D | ✗ | 11,619 | $\sim 20$ | 1.1M | 54,485 | ✗ | ✗ |
| GraspNet [19] | 6D | 3D | ✔ | 88 | ~400K | 1.2B | 97K | ✗ | ✗ |
| OCID-Grasp [53] | Rect. | RGB-D | ✔ | 89 | $\sim 7$ | 75K | 1,763 | ✗ | ✗ |
| MetaGraspNet [54] | 6D | 3D | ✔ | 82 | 1-5K | N/A | 217K | ✗ | ✗ |
| ACRONYM [55] | 6D | 3D | ✔ | 8,872 | 2,000 | 10.5M | N/A | ✗ | ✗ |
| Grasp-Anything [56] | Rect. | RGB | ✔ | 3M | 200 | 600M | 1M | ✗ | ✗ |
| ReasoingGrasp (ours) | 6D | RGB-D | ✔ | 64 | ~40K | ~99.3M | 97K | ✔ | ✔ |

# H Object and Part Statistics

| | Object Name | Part Name | 6D Grasps per Sample | Rect Grasps per Sample | Num. Scenes |
|---|---|---|---|---|---|
| 0 | Cracker Box | - | 10,971 | 4,672 | 35 |
| 1 | Tomato Soup Can | - | 16,625 | 8,284 | 34 |
| 5 | Banana | Stem, Flesh | 1,298 | 2,941 | 25 |
| 7 | Red Mug | Handle, Rim, Body | 3,363 | 2,045 | 25 |
| 8 | Power Drill | Handle, Chuck, Battery Pack, Body | 2,445 | 4,476 | 26 |
| 9 | Scissor | Handle, Blade | 548 | 574 | 22 |
| 11 | Strawberry | - | 130 | 903 | 24 |
| 14 | Peach | - | 960 | 1,922 | 25 |
| 15 | Pear | - | 948 | 2,069 | 26 |
| 17 | Orange | - | 581 | 1,466 | 32 |
| 18 | Knife | Handle, Blade | 1,975 | 1,526 | 29 |
| 20 | Red Screwdriver | Handle, Shaft | 1,191 | 1,910 | 29 |
| 21 | Racquetball | - | 443 | 573 | 6 |
| 22 | Blue Cup | Rim, Body | 3,960 | 2,770 | 29 |
| 36 | Daobao Wash Soap | Cap, Body | 2,594 | 1,744 | 25 |
| 37 | Nzskincare Mouth Rinse | Cap, Body | 1,806 | 1,786 | 24 |
| 38 | Daobao Sod | Cap, Body | 3,256 | 2,227 | 25 |
| 40 | Kispa Cleanser | Cap, Body | 931 | 1,654 | 25 |
| 41 | Darlie Toothpaste | Cap, Body | 442 | 1,181 | 25 |
| 43 | Baoke Marker | Cap, Body | 953 | 697 | 21 |
| 44 | Hosjam Toothpaste Pump | Pump, Body | 725 | 1,958 | 22 |
| 46 | Dish | Rim, Body | 8 | 22 | 25 |
| 48 | Camel | Legs, Head, Body | 96 | 663 | 25 |
| 51 | Elephant | Legs, Head, Body, Truck | 218 | 1,692 | 24 |
| 52 | Rhinocero | Legs, Head, Body, Horn | 152 | 1419 | 22 |
| 57 | Weiquan Chicken Bouillon | Cap, Body | 2,297 | 2,617 | 25 |
| 58 | Darlie Toothpaste Box | - | 5,526 | 1,646 | 25 |
| 60 | Black Mouse | - | 170 | 618 | 25 |
| 61 | Dabao Facewash | Cap, Body | 4,683 | 2,297 | 22 |
| 62 | Pantene Shampoo | Cap, Body | 502 | 1,175 | 26 |
| 63 | Head Shoulders Supreme | Cap, Body | 777 | 1,700 | 24 |
| 66 | Head Shoulders Care | Cap, Body | 1,338 | 1,269 | 24 |
| 70 | Tape | - | 4,465 | 1,859 | 25 |

**Table 5:** Selected Objects and Parts for the Training Split of the Dataset

# I  Hyperparameters

For the multi-modal LLM in the proposed model, we utilize the pre-trained LLaVA-7B-v0, which is derived from the large language model LLaMA-7B. As for the LoRA fine-tuning, we choose a rank $r = 64$. During the training, we use the batch size of $8$, and the learning rate is set to $5e - 4$, utilizing a cosine annealing schedule for adaptive learning rate adjustment. For the relative weight parameters in the loss function 1, we set $\lambda_t = 1$ and $\lambda_g = 1$ for the initial configuration, and starting from the third epoch, the parameter $\lambda_t$ is adjusted to $0$. For the train-test-split, we use $90\%$ of the first 100 scenes in GraspNet-1 billion [19] for training and $10\%$ of the first 100 scenes for testing. We sampled 50k from the original LLaVA Instruct 150K dataset [12] to mix with 100k grasping data to maintain the visual reasoning capabilities.

# J  Experiments on Visual Reasoning Ability.

To evaluate the visual reasoning capabilities of our model, we employ a GPT-4 judge to assess the quality of responses generated by our model. For each given image and question pair, both the original LLaVa model and our fine-tuned version provide answers related to the input image. We subsequently submit the questions, and answers from both LLaVA and our model, along with the ground-truth text descriptions, to a text-only GPT-4-based judge. This judge assesses the responses based on their helpfulness, relevance, accuracy, and level of detail in comparison to the ground-truth descriptions. GPT-4 then allocates an overall performance score ranging from 0 to 10, where higher scores denote superior performance. The image-question-answer sets used in this evaluation are randomly chosen from the LLaVa-Instruct-150k dataset [12] with multi-round conversations.

