# OpenReview forum: "Reasoning Grasping via Multimodal Large Language Model"
_robot-learning.org/CoRL/2024/Conference — CoRL 2024_

### Official Review · Reviewer_YHuz · 2024-07-15

**Originality:** 4
**Technical Quality:** 4
**Clarity Of Presentation:** 5
**Potential Impact:** 4
**Recommendation:** 4
**Confidence:** 5

**Review:**

## Strengths

1. The main contribution of the paper is both in a novel use of open-source VLMs for grasping with non-specific queries and in curating a new language-grasp dataset. This combination leads to an interesting grasp prediction system.

2. The two main baselines use CLIP instead of LLAVA and use bounding boxes from LLAVA instead of custom embeddings. Both of these baselines are very reasonable. The proposed method is shown to outperform these baselines in simulated and real-world experiments.

3. I like the analysis of possible scenarios with non-specific user queries in Appendix A. There are indeed many practical applications where the motivation of this paper makes a lot of sense.

## Weaknesses

1. The objects used in the experiments appear to be quite distinct from each other. As outlined in Appendix A, possible applications might include picking the clothes with the right color and pattern, or identifying “morning medicine”. I presume these tasks would require the model to distinguish similar looking objects, which may not be possible (or is at least untested) with the current architecture that averages the tokens that describe the object to be grasped.

2. The real world version of the proposed system requires a person to rank the top-3 grasps to achieve a reasonable success rate.

3. The discussion of the limitations of the method in the conclusion is insufficient. I would like to better understand the difficulty in selecting the right object vs the difficulty in generating an accurate grasp based on the VLM conditioning.

## Detailed comments

* On line 168, “customized variant of the GPT-4 model” would suggest that GPT-4 was fine-tuned, but I do not think that is the case?

* In Section 5.2, it is unclear if the part decomposition is already present in the dataset, if it is done by hand, or using a model.

* The two criteria for evaluating a grasp as successful described on line 227 might be too lenient (e.g. 25% IoU threshold is low).

**Quality Of The Limitations Section:**

2

**Questions For Rebuttal:**

The main reason for low real-world performance is stated as objects slipping from the gripper. But, the Franka robot is more than capable of lifting these objects. Are the failures due to inaccuracy of the (x, y, theta) prediction from the grasp network or from the grasp height prediction, which I presume is chosen using a heuristic?

**Robotics Focus:**

4

**Summary Of Paper:**

This submission describes a novel grasp prediction system based on open-source vision language models and grasp prediction networks. The system includes LLaVA, a vision language model, that is fine-tuned using low-rank approximation to produce specific tokens that identify objects. These tokens are then passed into a 2D grasp prediction network. The main claim of this paper is that their model is able to follow vague instructions, such as “I am hungry and I would like to eat”. This is achieved in part by using a strong VLM and in part by creating a new dataset that has specific and vague instructions annotated by GPT-4 and checked by humans.

**Summary Of Recommendation:**

I think the proposed system that fine-tunes open-source models to convert vague queries into grasps is interesting and significant. The baselines are sufficient, but the real-world robot performance is quite low. All of the evaluated methods possibly struggle with predicting accurate grasps, which is strange considering the amount of strong grasping models. Nevertheless, my opinion is somewhere between weak and strong accept.

---

### Official Review · Reviewer_NKtc · 2024-07-19

**Originality:** 3
**Technical Quality:** 3
**Clarity Of Presentation:** 4
**Potential Impact:** 3
**Recommendation:** 2
**Confidence:** 4

**Review:**

## Strengths
- Overall, the paper is well-structured, making it easy to understand the entire study.
- The task definition is clearly written, which helps in comprehending the issues addressed in this paper. Figure 1 is also well-organized, showing the differences in experimental conditions clearly, aiding in understanding.
- Executing tasks through implicit instructions is extremely important for real-world applications, and the experiments using these instructions are highly valuable. Indeed, in experiments using implicit instructions, the proposed method achieves a higher task success rate than baseline methods. Additionally, proposing a dataset that includes these instructions is a significant contribution to the field. Particularly, Tables 4 and 5 are valuable references when considering the dataset's contributions.
- The submitted videos are well-organized and greatly assist in understanding the paper. They include videos of real-world experiments, providing good material for validating the method's effectiveness.


## Weaknesses
- In Table 2, the performance of Reasoning Grasping decreases overall when using depth images. There is insufficient discussion on the reasons for this decline.
- Compared to GR-ConvNet, the technical advancements are limited. Therefore, the technical contribution of Reasoning Grasping is very limited.
- There is no ablation study. One of the most important elements of Reasoning Grasping is the introduction of the special token.
Additional experiments, such as ablation studies without the special token or without using LoRA, are needed to investigate the contribution of the special token.
- For the "LLaVA -> GR-ConvNet Baseline", I understand that the purpose of using LLaVA is to obtain bounding boxes for candidate objects. If so, instead of LLaVA, Open-vocabulary object detectors such as Grounding DINO [1] or UNINEXT [2] should have been used. The authors should conduct experiments with a pipeline that combines these Open-vocabulary object detectors with GR-ConvNet as the baseline method.


[1] Liu, S., Zeng, Z., Ren, T., Li, F., Zhang, H., Yang, J., Zhang, L. Grounding DINO: Marrying DINO with Grounded Pre-Training for Open-Set Object Detection. arXiv preprint arXiv:2303.05499, 2023.
[2] Yan, B., Jiang, Y., Wu, J., Wang, D., Luo, P., Yuan, Z., & Lu, H. Universal Instance Perception as Object Discovery and Retrieval. In Proceedings of the IEEE/CVF Conference on Computer Vision and Pattern Recognition, pages 15325-15336, 2023.

## Minor Comments
- In line 187, only the numbers have commas. Commas should be added to other numbers as well.
- In the top right diagram of Fig. 3, there is an extra double quotation mark. Additionally, in Fig. 3, the instructions are very difficult to see because they are small and in white text.

**Quality Of The Limitations Section:**

3

**Questions For Rebuttal:**

Please review the above weaknesses. Below are additional questions.
1. How are the rotation and width specifically determined? How should the maps related to them in Fig. 2 be interpreted?
1. For a quantitative comparison of Ex/Implicit instructions, statistical data on the instructions included in the dataset should be provided. What are the average sentence lengths and vocabulary sizes for both Ex/Implicit instructions? Additionally, on average, how many words constitute the special token?

**Robotics Focus:**

4

**Summary Of Paper:**

This study proposes a grasp pose estimation method called Reasoning Grasping. The novelty of this method lies in using MLLM to predict affordance ('special token'), which is expected to improve the understanding of which objects should be focused on. Additionally, a new dataset for natural language-based grasp pose estimation is constructed and proposed. Experimental results show that the proposed method can estimate grasp poses more accurately than baseline methods.

**Summary Of Recommendation:**

The technical contribution of the special token is indeed recognized. Furthermore, the construction of a dataset for grasp pose estimation that includes instructions and affordances expressed in natural language, which did not exist in previous datasets, is a significant contribution to the field. However, as mentioned above, there are several concerns with this paper, including the lack of ablation studies and limitations in novelty. These issues should be resolved before the paper is published. Therefore, my decision is "weak reject."

---

### Official Review · Reviewer_ucUd · 2024-07-21
**The paper defines novel task and proposes extensive dataset with good performance**

**Originality:** 4
**Technical Quality:** 3
**Clarity Of Presentation:** 4
**Potential Impact:** 3
**Recommendation:** 3
**Confidence:** 3

**Review:**

Strengths:
* The paper is well motivated by clearly showing the importance for robots to interpret implicit instruction and the definition of reasoning grasping is novel. The paper is written clearly in detail. Figures are clear.
* The paper creates a reasoning grasping dataset based on the GraspNet-1 billion dataset, which contains 1730 instruction pairs and is a valuable contribution.
* The paper examines each part of the model, and clearly outperforms the baselines.
* The paper conducts real-world experiments, which demonstrates its real-world applicability.

Weaknesses:
•	The model has multiple stages, which could result in some latency if applied to real-world applications.
•	The paper tests and reports the result of text generation and reasoning grasping separately. It would be good to report the end-to-end performance as well.
•	The model relies on static images input, which could have difficulty to deploy in cluttered environments where objects are occluded or partially-occluded.

**Quality Of The Limitations Section:**

3

**Questions For Rebuttal:**

* Include user study and evaluate user satisfaction could demonstrate the effectiveness of the model with real-world user-created implicit instruction.
* Include the model end-to-end performance as well.

**Robotics Focus:**

4

**Summary Of Paper:**

The paper introduces a novel task, reasoning grasping, that direct robot grasping with implicit instructions, create a reasoning grasping dataset, and propose a multi-modal LLM model to solve this task. In addition, the paper also performs real-world experiment to demonstrate the model’s effectiveness.

**Summary Of Recommendation:**

The paper defines an interesting and novel task: reasoning grasping. The dataset includes 1730 implicit instructions. The proposed model has good performance and could be applied to real world.

---

### Author Rebuttal · Authors · 2024-08-12

We attached the updated paper.

---

### Decision · Program_Chairs · 2024-09-04

**Decision:**

Accept

**Comment:**

This paper proposes a novel grasping system that combines a VLM with a grasp prediction network. The VLM takes in a natural language query, and outputs a special token that is used to condition the grasp prediction network.
This way, more complex grasp queries in natural language can be achieved compared to other methods.

Despite one reviewer questioning the technical novelty of the paper, paper contains enough original ideas. The empirical evaluation is also sound.

The authors should make their dataset publicly available.